# Processing Radargrams to Obtain Resistivity Sections

**Lucía Arévalo-Lomas** [1] , **Bárbara Biosca** [1] , **David Paredes-Palacios** [2] **and Jesús Díaz-Curiel** [1,*]

1   Department of Energy and Fuels, School of Mines and Energy, Universidad Politécnica de Madrid, C/Ríos Rosas 21, 28003 Madrid, Spain; lucia.arevalo@upm.es (L.A.-L.); barbara.biosca@upm.es (B.B.)
2   Department of Geological and Mining Engineering, School of Mines and Energy, Universidad Politécnica de Madrid, C/Ríos Rosas 21, 28003 Madrid, Spain; david.paredes@upm.es
*   Correspondence: j.diazcuriel@upm.es

**Abstract:** Ground penetrating radar (GPR) is routinely used to locate the isolated elements that produce reflection hyperbolas in radargrams. However, we propose a method in this study for locating the various interfaces appearing in a medium by studying the signal attenuation to obtain resistivity sections. GPR signal decay has a strong relationship with the electromagnetic properties of the medium, particularly the electrical resistivity and permittivity. To assign values of resistivity to different layers, a relationship between the attenuation coefficient and the above parameters must be used. Moreover, there are geometric effects that affect the energy loss and, therefore, the signal amplitude, that are jointly considered for the elimination of such effects before calculating the attenuation coefficient. An envelope function of the traces previously corrected for geometric effects was created to detect interfaces in the medium and generate a local decay curve and radargram zonation. Two relationships are necessary for obtaining the resistivity values from signal decay: first, a relationship between the resistivity and relative permittivity of the medium; and second, a relationship between the attenuation coefficient and resistivity. A resistivity section obtained from the GPR data is shown with an electrical tomography section at the same location.

**Keywords:** ground penetrating radar (GPR); signal attenuation; resistivity; permittivity; filter design; zonation

## 1. Introduction

One widely accepted subsurface characterization parameter obtained through geophysical prospecting methods is the electrical resistivity, which is related to other characteristics such as the compactness, moisture content, or the type of fluid filling pores; all of these are in high demand in many areas of engineering. In this study, we propose a methodology to extract that parameter from the measured profiles of ground penetrating radar (GPR).

GPR is commonly used to locate isolated reflectors (natural or anthropogenic) or the interface morphology with strongly contrasting electrical properties [1–3] in the study of overlay-bedrock, ice-substrate, or moraine rock. It is also used for the detection of groundwater levels or contaminant plumes. In the last decade, police investigations have intensively used GPR to locate buried bodies and rescue workers after natural disasters or accidents [4].

In recent years, studies have focused on data processing to obtain radargrams in which existing anomalies can be easily identified [5–9]. The propagation of electromagnetic fields, which is well known from Maxwell's equations [10], depends on the sources that generate them and several constants that depend on the medium in which they propagate. These latter characteristics cause signal attenuation in radar waves as they propagate through the medium.

The existing literature notes different mechanisms to analyze the GPR attenuation, such as dispersion [11], which does not significantly affect the ground on which GPR is

usually employed, and absorption and scattering [12,13]. Many works have shown the complexity of the estimation of GPR signal attenuation [14–16], but in this study a simplified alternative approach is presented. In general, spherical divergence and absorption due to the electromagnetic properties of the medium are the most important mechanisms [17].

In many radargrams, both obtained by us and in those shown in many published works, there are areas where the attenuation of the GPR signal is clearly differentiated. To understand the different attenuations shown in the analyzed radargrams, several processing steps were programed and integrated into a set of algorithms implemented in MATLAB ©, which also enables the possibility of graphically representing the obtained results. Other authors have used this language to create several GPR signal processes; however, in most cases, it has been used for simulation and modeling [14,18,19].

In this paper, a methodology that allows additional and unusual information to be extracted from radargrams beyond the most common data interpretations of this technique is presented. Specifically, resistivity sections were obtained as a function of the depth of the investigated soil after subjecting the original signals to various processing steps. First, the effects independent of the medium that caused some attenuation of the GPR waves were eliminated. The decay of the resulting signal depends only on the characteristics of the medium. Then, after modeling the emitted pulse, an envelope function of the signals, whose decay, defined by the attenuation coefficient ($\alpha$), depends only on the characteristics of the medium, was developed. Empirical relationships between the relative permittivity of granular porous media and resistivity, and between the attenuation coefficients and resistivity, were established. As an example, we depict a resistivity section from the study of the attenuation of GPR traces ($\rho_{\text{GPR}}(\alpha)$), wherein different layers can be distinguished, and the results are compared with an electrical tomography section obtained in the same area.

In short, starting from previous concepts in the field of electromagnetic waves, the fundamental contribution of this work is to provide a first approximation of the resistivity distribution in the subsoil from GPR data, using simple expressions that first relate the electrical permittivity of a medium with the resistivity, and then the resistivity of the medium with the attenuation coefficient of electromagnetic signals.

*Background*

The theoretical bases needed for processing radargrams range from the elimination of the effects derived from the measurement system to corrections of the measured signals, including the effect of spherical divergence [20], in addition to the study of the effects of energy loss on signals of the electromagnetic properties of the medium (attenuation). The first correction of the measured signal involves the elimination of the DC component or the tendency due to the recording rate [21]. To develop the signal analysis, the latter should be conducted first.

One of the first problems typically presented by GPR signals involves a shift in all signals from the zero value of the amplitude [22]. This is known as the DC offset of the traces, and it occurs when the average or median value of the data is not close to zero; therefore, the trace is not centered. To rectify this shift, filters (i.e., mathematical functions) are applied over the signals.

The DC offset can be corrected by applying a filter based on the calculation of the median value of the data, whereas other methods such as averaging can also be applied [21]. The DC offset can also be corrected by applying a "running average" filter [23–25]. Other authors have referred to this type of processing (offset removal filter) as a DEWOW filter [26,27]. This DC offset removal filter is used by most authors [25,28–33], and they all agree on the importance of removing this effect at the beginning of data processing in order to avoid negatively influencing the information provided by the signal amplitudes in the subsequent stages. Moreover, the median value has been used for various purposes by some authors [21,34,35] as it is unaffected by extreme values.

Another feature of GPR signals is their amplitude loss due to geometric effects. These are mainly caused by the following: spherical divergence [20], wherein the signal attenuates

as the square of the distance traveled according to the law of flux conservation; and the dipole effect, wherein the generated dipole field is zero in the plane of the antennas and maximum in the perpendicular direction. Although many studies on the dipole field of GPR antennas have investigated the fields emitted by high-frequency antennas [36], which are based on the antenna's direction and polarization [37], these fields may vary depending on the heterogeneity of the medium through which the signal propagates. The author of [38] attempted to solve the problem of dipole antenna radiation in the presence of a stratified medium by decomposing the total wave field into its electrical and magnetic components. An example of such a near-surface formulation for horizontal dipoles and a comparison with the geometrical optics approximation can be found in [39]. Conversely, article [40] presents the formulation for a horizontal dipole in the presence of a conducting half-space.

Regarding the absorption of a part of this radiation during the propagation of waves in the medium, the author of [41] stated that the penetration of very high frequency electromagnetic radiation into rocks was considerably low, thereby ruling out its use as a practical method of geophysical prospecting. However, applications in surface prospecting are feasible, such that information about the characteristics of the medium can be extracted by studying the behavior of the signals.

These electromagnetic parameters are related to signal decay through the attenuation or absorption coefficient $\alpha$, which is extracted from the development of Maxwell's theory concerning the attenuation of electromagnetic fields; this has been reported by numerous authors [42–46].

Generally, the signal is considered to decay with time (or depth) according to an exponential function of the following type [10]:

$$A(t,z) = A_0\,(t) \cdot e^{-\alpha z},\tag{1}$$

where $A_0$ is the initial signal amplitude, t is the time of arrival of the signal, and z is the signal depth. The attenuation coefficient [42] is given by Equation (2):

$$\alpha = \frac{\omega}{c}\left[\left(\frac{\mu_r \varepsilon_r}{2}\right)\left\{\left(1 + P^2\right)^{\frac{1}{2}} - 1\right\}\right]^{\frac{1}{2}}\tag{2}$$

where $\omega$ is the angular frequency of the antenna; c is the velocity of electromagnetic waves (speed of light) in vacuum ($3 \times 10^8$ m/s); $\mu_r$ is the relative magnetic permeability of the ground; $\varepsilon_r$ is the relative permittivity of the medium; and $P$ is the loss tangent of the angle, expressed as $P = \sigma/\omega\varepsilon$.

Among the three electromagnetic parameters shown in Equation (2) (the relative permittivity, electrical conductivity, and magnetic permeability), the latter can be considered to be invariable for all grounds usually studied using the GPR technique, which can be considered paramagnetic; it can therefore be attributed to the relative magnetic permeability of the unit value [47]. Therefore, in Equation (2), the attenuation coefficient is only a function of the resistivity and relative permittivity as the other elements of the equation are known for a given antenna frequency.

The relationship in Equation (2) shows how the signals attenuate the higher conductivity of the medium through which they pass. Thus, the study of wave attenuation can provide important information for various fields, such as detecting the movement of contaminant plumes in the ground [15,48,49], risk assessment [50,51], and saline tracer flow modeling [52,53], among others. The attenuation coefficient can be considered to be an intrinsic electromagnetic property that is a function of the conductivity, dielectric constant of the soil, and fluid filling pores. Furthermore, because the resistivity of the medium is highly dependent on the fluid within the pores, some authors have considered the moisture content as the factor governing the depth of investigation [43].

The velocity of electromagnetic waves in the studied medium is directly related to the depth and can be expressed as [42]:

$$v = \sqrt{2}\left(\frac{\omega}{\mu}\right)^{1/2}\left[\left(\varepsilon^2\omega^2 + \sigma^2\right)^{1/2} + \omega\varepsilon\right]^{-1/2} \tag{3}$$

Thus, the wave velocity in the medium is defined, and the depths at which the different heterogeneities of the medium are found can be determined. The investigation depth of GPR is limited by all of the aforementioned attenuation phenomena; these include geometric criteria including the exponential decay with depth that is independent of frequency [54] and, conversely, the antenna frequency and medium resistivity. Summarizing, the GPR wave propagation is governed by the product of a real exponential function depending mainly on the attenuation coefficient and an imaginary exponential function, which includes the frequency of the wave related with several variables interdependent between them.

Regarding the calculation of velocities, some authors [55] consider that the velocity is constant up to the first reflector. This conclusion was based on studies with antennas wherein the separation between the transmitter and receiver was variable.

According to the authors' perspective, conducting vertical electrical sounding or time-domain electromagnetic sounding in the study area was the best way to obtain an approximate resistivity value of the ground. Regarding GPR, two main energy losses of GPR signals may be considered: First, a more well-known energy loss in reflection/transmission at existing interfaces between media with different electromagnetic properties, particularly resistivity. A strong contrast in this will generate strong reflections and strong signal attenuation. Second, the energy loss inside each medium due to attenuation because of its non-infinite resistivity, which is the effect analyzed in this study. One main advantage of this methodology is that the attenuation calculation is mainly dependent on the resistivity values. Therefore, all decay that can be measured in the resulting trace is attributed to the absorption effects of the medium and to reflection and transmission effects at the existing interfaces due to the medium's electromagnetic properties.

## 2. Materials and Methods

The methodology presented in this work consists of several steps grouped into two main stages. In the first, the aim is to correct the decay due to purely geometrical considerations and independent of the electromagnetic properties of the medium from the GPR signal. In the second stage, considering that the behavior of the signal resulting from the first stage is due solely to the electromagnetic properties of the medium, an envelope function for each trace is determined, from which the attenuation coefficients of the different layers traversed by the GPR signal are computed along each trace. These attenuation values are related to the resistivity values using the equation proposed in this study.

### 2.1. Signal Preprocessing

Before analyzing the signal attenuation, preprocessing was conducted to ensure the accuracy of the subsequent analysis. Because subtracting a function with depth will produce different offsets with time along the trace (but not different with the frequency content), in this study a single value is subtracted for each trace, so that the amplitude and frequency of the signal does not change with time. A simple filter was developed to eliminate the DC offset with which we calculated the median of all the amplitude values of each trace after the initial strong undulations caused by the first reflection (i.e., from the sample for which the variance of two wavelengths is less than 1/25 times the maximum variance). This calculated value was subtracted from all samples of the studied trace.

Once the DC offset was removed, the radargrams were represented prior to processing using the logarithms of the amplitudes in a way similar to that of article [56]. As the decay is exponential, this visualization gives greater significance to the smaller amplitudes corresponding to the last arrival times. Figure 1 shows an example of radargrams before

and after DC correction on a logarithmic scale. The figure shows a significant difference in the case of the example presenting the corrected DC, and alternating positive and negative amplitudes are observed. Figure 1b shows a radargram with a severe so-called ringing, generally assigned to the contrast of physical properties between air and the topographic surface, or another shallow interface. Although one of the conventional processes aims to eliminate this ringing, this study focuses on the analysis of the decay of this signal that apparently contains a single frequency. Thus, although the application of a radon or f-k filter would eliminate the ringing seen in Figure 1b, producing different local amplitude changes in the radargram as a function of the wavenumber, it has not been applied to maintain the original wavelets of the propagation of the emitted signal.

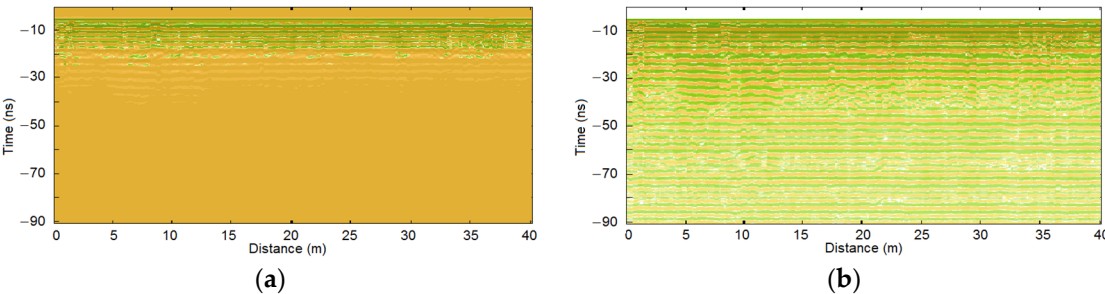

(**a**)    (**b**)

**Figure 1.** Representation of the logarithm of data: (**a**) before and (**b**) after applying the DC filter.

## 2.2. Removing the Geometric Effects

The next step consists of subtracting the effects from the signal generated by factors causing energy loss, and therefore affecting the amplitude. These are not attributed to the characteristics of the ground through which the waves propagate. These effects are defined as geometric effects, among which only two have been considered in this work: the spherical divergence and dipole effect.

The geometric spreading of waves can be considered from two perspectives. First, the conservation of the energy flux of a wave, known as spherical divergence attenuation, results from the propagation of the wavefront through any medium and the loss of amplitude as it moves further away from the emitting source [17,20]. With the used antenna, where the two dipoles are parallel between them and to the surface, considering both energy and direction, only the main lobe is analyzed as having the greater effect on the wave propagation. This effect is purely geometric and independent of frequency, and is inversely proportional to the square of the distance to the transmitting antenna (Figure 2), which is based on flux conservation. In this sense, the expressions used for the attenuation of radar waves maintain parallelism with those used in reflection seismics because of the similarity between the appearances of both types of signals. Many authors have reported this similarity [57–61]; thus, these studies on the attenuation of seismic waves can be used to process GPR signals.

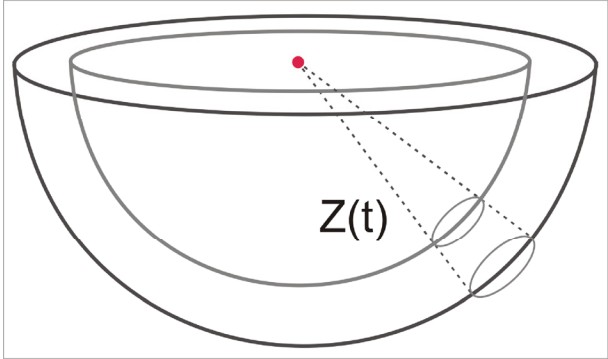

**Figure 2.** Explanation scheme of spherical divergence.

However, we must also consider the fact that the signal was emitted from a dipole source. This causes the signal to be zero in the plane of the antennas and maximum in the perpendicular direction. This can be considered to coincide with the vertical plane at a certain depth, when the angle between the vertical and field strength vector is almost zero (Figure 3), which coincides with the direction of the emitted signal beam. Therefore, we used the angle *θ* formed by the field strength vector and antenna axis, which is a critical factor [62].

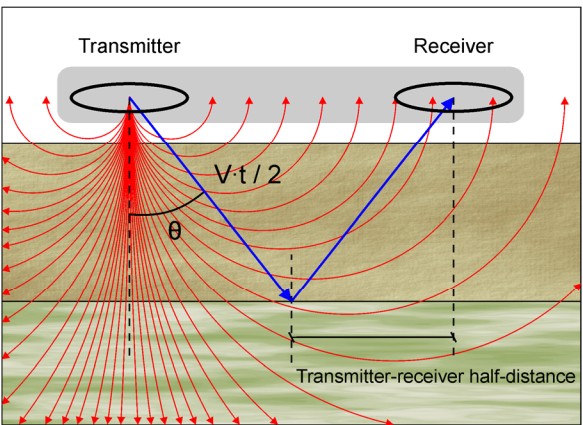

**Figure 3.** Illustrative sketch of the dipolar effect, which is zero in the plane of the antennas and maximum in the perpendicular direction.

To consider the above two effects together, we created a propagation function, as expressed by Equation (4), which considers both effects. In this expression, a constant K is included, which depends on the characteristics of the used antenna, and especially on the distance between the transmitter–receiver dipoles. It is not critical for determining the attenuation and has been obtained as an empirical adjustment constant, determined after analysis of much data, to be 1000 for the 500 MHz antenna:

$$Propagation\ function = \text{K} \cdot \frac{\cos\theta}{d^2} \tag{4}$$

On analyzing the expression of the propagation function, the cosine in the numerator represents the dipole effect, which causes the function to exhibit an increasing trend at the beginning, such that it approaches the behavior of the traces. Furthermore, the losses due to the conservation effect of the flux through a progressively increasing surface are reflected in the denominator of this formula by the factor $d^2$ (square of the traveled distance, v·t/2). The signal amplitude decreases by a factor equal to the square of the distance as the distance traveled increases. Thus, as shown in Figure 4, this function fits the trace being processed, implying that the generated propagation function reflects the GPR signal behavior.

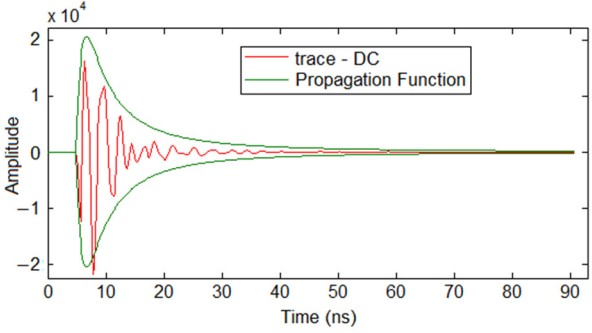

**Figure 4.** Propagation function (Equation (4)) applied to an example trace.

To undo the geometric effects, each trace from the radargram must be multiplied by the inverse of the propagation function. This results in traces of the type shown in Figure 5, wherein the initial amplitudes decreased more than the final amplitudes; thus, the ratio between the two is much smaller. This processing step should be perceived as a simulation indicating that the geometric effects do not affect the signals. Therefore, all decay that can be measured in the resulting trace is attributed to the absorption effects of the medium due to its electromagnetic properties.

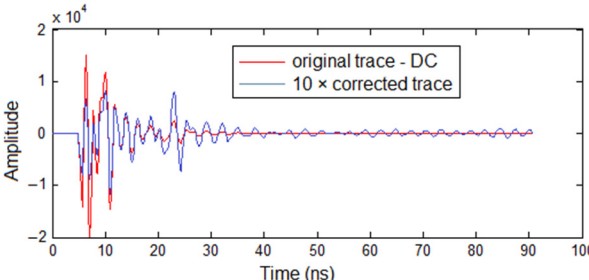

**Figure 5.** Example of an original trace (red) and the same trace corrected by the propagation function—geometric effects (blue). Corrected amplitudes have been multiplied by 10 for better visualization.

### 2.3. Calculating the Depths

The attenuation to be calculated must be a function of depth. Thus, it is necessary to generate a depth vector from the time vector obtained through GPR by considering the separation between the transmitter and receiver and the elevation of the antenna, in addition to the different propagation velocities of the waves in the air and the ground.

Considering the operation of the equipment, the first arrival detected by the receiver corresponds to the airwave. The airwave travels through the air with a velocity practically equal to the speed of light in vacuum, whose value is $3 \times 10^8$ m/s.

Assuming that the distance between the transmitter and receiver of the equipment used in this study is 0.18 m for the 500 MHz antenna, we calculated the time at which the airwave arrives. At subsequent times, the reflections start to arrive, first from the air–soil interface and then from the interfaces found in the ground or from any element in the subsoil with different electromagnetic properties.

To correctly calculate the depths, we consider the part of the wave that travels through the air before entering the ground and just before it is received by the detector at a distance equivalent to the height of the antenna above the ground. Considering several geometrical considerations (Figure 6), we obtained the following expression relating both parameters:

$$Z(t) = v_2 \cdot t - K_z \tag{5}$$

where $v_2$ is the velocity of the electromagnetic waves in the medium and $K_Z$ is a constant with a value of 0.03. Initially, this velocity is unknown but a velocity of 0.08 m/ns is used, which is equivalent to an approximate resistivity of 100 $\Omega \cdot$m. This is a valid average value for the type of medium studied in this work.

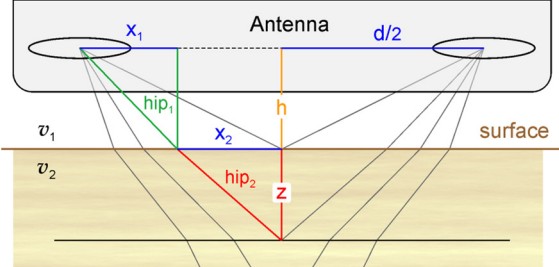

**Figure 6.** Geometrical considerations for the calculation of $Z(t)$.

### 2.4. Programming and Defining the Envelope

The signal decay was studied by analyzing the trend of the traces after correction. For this purpose, an envelope function was generated to represent this trend, which enabled us to calculate the values of the attenuation coefficient representative of the traces. This function was obtained from the relative maxima of the absolute amplitude values. It is an envelope function that, in principle, would be perfectly defined as the curve passing through all the relative maxima of the record if each of them was always smaller than the one immediately preceding it.

However, relative maximums higher than previous ones appear along the traces in the recording; namely, the intermediate amplitude increases, as opposed to the continuous decay that would be expected in a homogeneous medium due to the presence of media (reflectors) of different electromagnetic properties. It is reasonable to expect that, due to both the electronics of the emitting circuit and the matching (resonant circuit) of the antenna system to the emitted frequency, the emitted pulse has a characteristic shape, as shown in Figure 7. In particular, the maximum emission value at the antenna does not occur instantaneously but over a few nanoseconds. After analyzing the obtained traces, we estimated that, in the case of the used equipment, this time was approximately 5 ns. This fact justifies both the gradation of the initial increase in the amplitude values, and the small intermediate increases throughout the recording every time the signal encounters an interface in the ground that causes reflection.

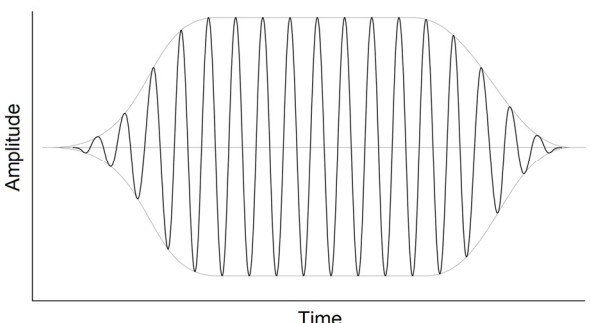

**Figure 7.** Characteristic shape of the emitted pulse.

With the GPR emission system considered in this study, the signal is not assumed to consist of a single wavelet that is repeated at intervals given by the inverse of the antenna frequency, but pulses containing this frequency, with an interval of duration prefixed in each case, which are repeated to obtain an adequate stacking to improve the signal-to-noise ratio.

Therefore, in the generated envelope function, certain criteria were established to disregard some of these maxima that correspond to the rise of the emitted pulse and not to the behavior of the signal in the presence of reflecting layers. Thus, the successions of stepped maxima coinciding with the shape of the emitted pulse were eliminated, considering only the last successions. Finally, we acquired the appearance of an envelope that is unaffected by the initial shape of the emitted pulse. Figure 8 shows an example of the result of the envelope function, wherein increases in the intermediate amplitude values can be observed, although the overall behavior of the curve shows a decreasing trend.

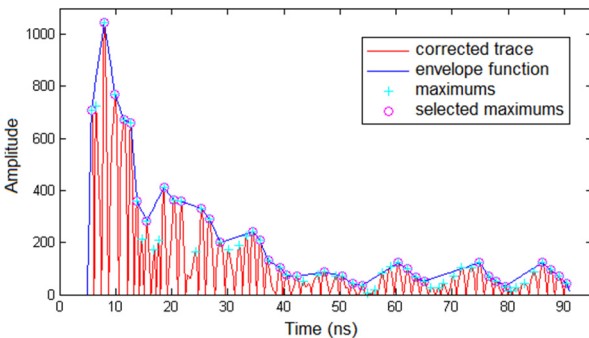

**Figure 8.** Envelope function fitted to the absolute values of the amplitudes.

*2.5. Calculating the Attenuation along the Envelope*

If an exponential curve is fitted to the entire defined envelope function, the overall attenuation coefficient is obtained for each trace. However, for the localization of different layers, it is necessary to consider how the decay varies from the shallowest reflections to the latest recording times. For this purpose, successive sampling windows with different numbers of data, and different overlaps between them, were fitted to $I_w = I_{0w} \cdot e^{-\alpha \cdot t}$, where w denotes the successive windows, $I_{0w}$ the ordinate at the origin for each window, and $\alpha$ the attenuation coefficient along each window. After testing different wide windows and overlaps, a window of 100 data points with an overlap of 20 was considered.

Thus, a "stepped curve" representing the different attenuation coefficients of each trace was obtained, as shown in Figure 9 for two isolated traces.

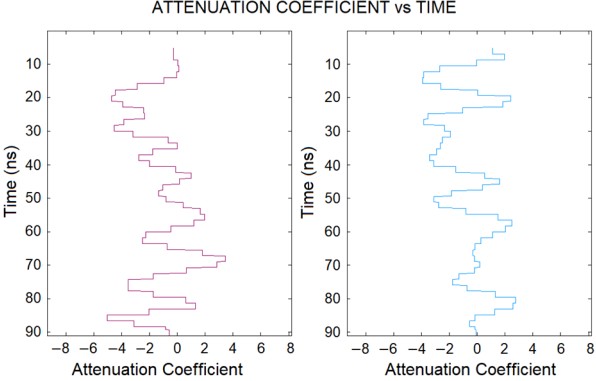

**Figure 9.** Stepped curves of attenuation coefficients obtained for two isolated traces.

Once the stepped curves of all traces were obtained, they were filtered to eliminate the result of the discretization of the attenuation, and their critical points were determined by calculating the first and second derivatives. The subsequent step was to jointly analyze the location of these critical points in relation to the shape of the envelope curve. The top of each new layer is located at the point where the amplitude of the envelope starts to increase, whereas the bottom coincides with the minima, which subsequently grows because of the appearance of a new interface.

Having defined the existing layers in each trace, and similar to how attenuation coefficients were calculated with 100 data windows, we calculated the attenuation coefficients for the defined layers. For this purpose, we considered the intervals defined by the critical points in the smoothed curve of the initial attenuation discretization. In this case, only the decreasing parts of the envelope were considered; namely, those showing an actual decay in the signal. Figure 10 shows an envelope wherein the relationship between the behavior of the envelope function and the determined critical points in the smoothed stepped curve can be seen, along with the various layers for that particular trace.

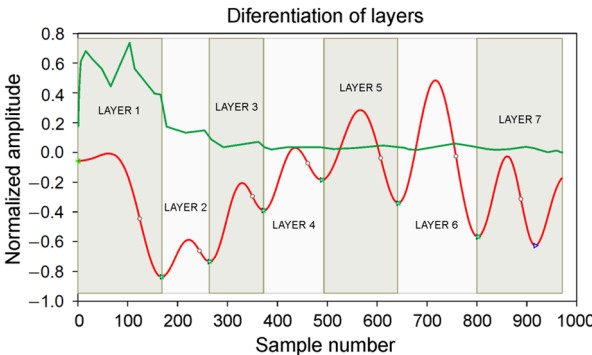

**Figure 10.** Joint representation of the envelope (green) with the smoothed steps (red). The ○ symbols indicate critical points showing the differentiation of layers.

Thus, the variation in the attenuation coefficient with time or depth can be obtained for all traces of the radargram (Figure 11 shows the results of two example traces). Considering that this information is available for each trace constituting the radargram, all traces can be concatenated to obtain sections of the attenuation coefficient as a function of time or depth.

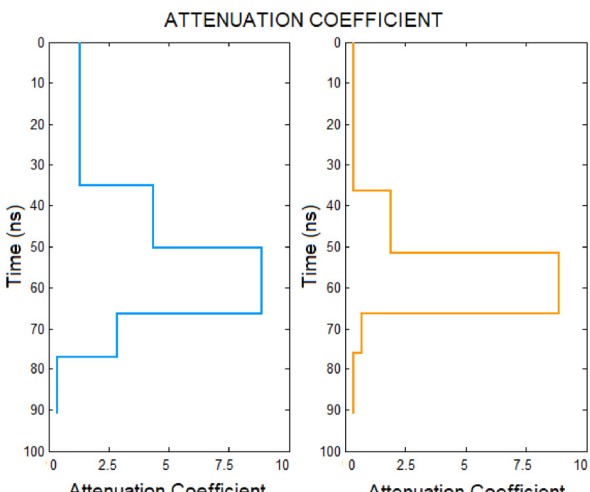

**Figure 11.** Representation of the attenuation coefficient as a function of time in two example traces.

### 2.6. Transformation of Attenuations into Resistivities

The absorption coefficient of the medium provides information about its electrical conductivity values. Some authors have developed experimental relationships [1] that relate both parameters using the following expression:

$$\sigma = \frac{\alpha \sqrt{\varepsilon_r}}{194.5} \tag{6}$$

To further simplify the attenuation coefficient equation extracted from Maxwell's equations (Equation (2)), an experimental relationship (Equation (7)) was developed from the data extracted from the literature [35,63]. This expression relates the relative permittivity to the resistivity of the same medium for the final calculation of resistivity values. These values are plotted in Figure 12 and are approximated to the potential function of Equation (7), which has a regression coefficient of 0.74. The correlation between electrical resistivity and relative permittivity of the geological medium depends on the mineral grain (when they are as conductive as clays), and on the type and content of ions in the formation fluid. Thus, the relationship obtained between them may be initially considered to be a

strong assumption. However, the obtained $R^2$ value allows a certain degree of confidence to be assigned, which is at least similar to that assigned in the cited references [35,63].

$$\varepsilon_r(\rho) = \frac{44}{\rho^{1/4}} \tag{7}$$

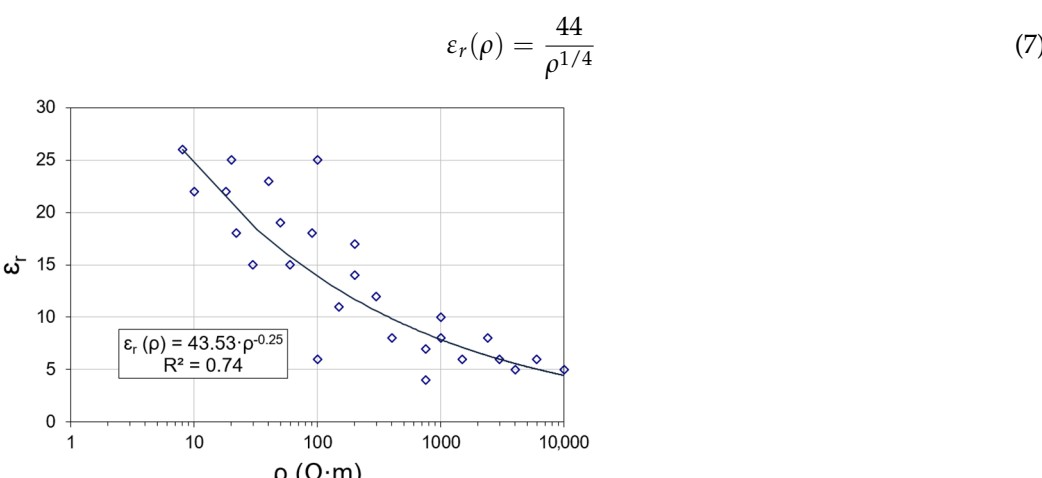

**Figure 12.** ◊ Relative permittivity versus resistivity values in different media and the least squares approximation curve.

Substituting Equation (7) into Equation (2), the expression of the attenuation coefficient is simplified as a function of the frequency and electromagnetic parameters of the medium. Thus, by assigning values for the resistivity, the corresponding values of the attenuation coefficient are obtained. By fitting the pairs of values obtained (Figure 13) to a curve, a potential expression was obtained from the resistivity as a function of the attenuation coefficient (Equation (8)), which presents a regression coefficient equal to 1 as Equation (8) is a potential function.

$$\rho(\alpha) = \frac{45}{\alpha^{1.15}} \tag{8}$$

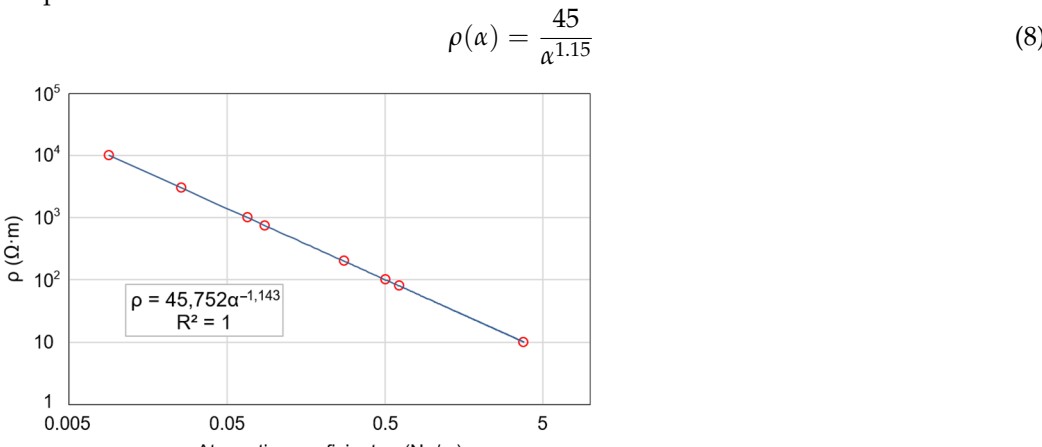

**Figure 13.** Resistivity values and attenuation coefficient, and least squares approximation curve.

### 2.7. Case Study

The methodology was applied to data obtained in a study area in the province of Guadalajara (Spain). The geology of the area consists mainly of Middle Triassic materials, specifically detrital materials such as sandstones with intercalations of silts and clays. The petrological study of the sandstones described in the literature indicates a quartz composition between 30 and 70 percent; potassium feldspar between 10 and 25 percent; and ferruginous cement, which can reach values of up to 15 percent. Dolomitic cement can occasionally reach up to 40 percent.

Regarding the equipment used for data acquisition, GPR data were acquired using the MALA-RAMAC 500 MHz antenna with the ProEx unit, with a trig interval (distance between traces) of 0.1 m and a sampling time window of 0.176 ns. The electrical prospecting

equipment used to measure the ERT profile was the Terrameter SAS 4000 from Guideline Geo-ABEM (Stockholm, Sweden). The profile was measured with a dipole–dipole array with electrode spacing of 1 m and 7 survey levels. The inversion of the profiles was carried out with the Res2Dinv Version 3.57 software from Geotomo© (Gelugor, Penang, Malaysia).

## 3. Results

After obtaining the attenuation coefficient as a function of time for each trace (Figure 11), to obtain attenuation coefficient sections, all traces were plotted together and isolines were generated with different colors depending on the value of the attenuation coefficient. Sections of the smoothed attenuation coefficient values were obtained depending on the time of arrival of the waves at the surface, as shown in Figure 14.

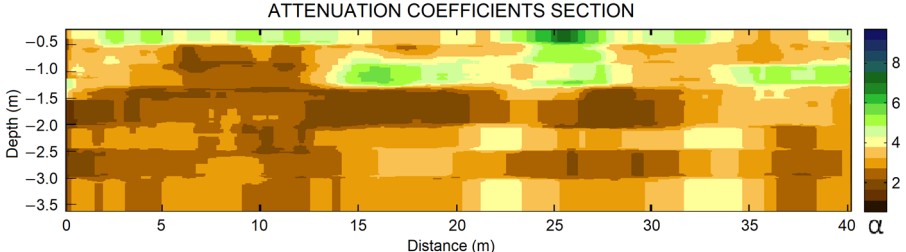

**Figure 14.** Example of a section showing the variation in attenuation coefficient values obtained from GPR profile.

The most favorable representation for the resistivity values obtained was in the form of smoothed sections to facilitate subsequent interpretation. This smoothing is conducted to obtain a more easily interpretable representation; thus, horizontal filtering is applied with windows of 16 datapoints to provide lateral continuity to the localized layers while maintaining the original resistivity values. Accordingly, sections such as those in Figure 15 were obtained. The attenuation coefficient and resistivity values related by Equation (8) are jointly represented. As shown, the low attenuation coefficient values coincide with high resistivity values and vice versa. The red and blue colors indicate higher and lower resistivity values, respectively.

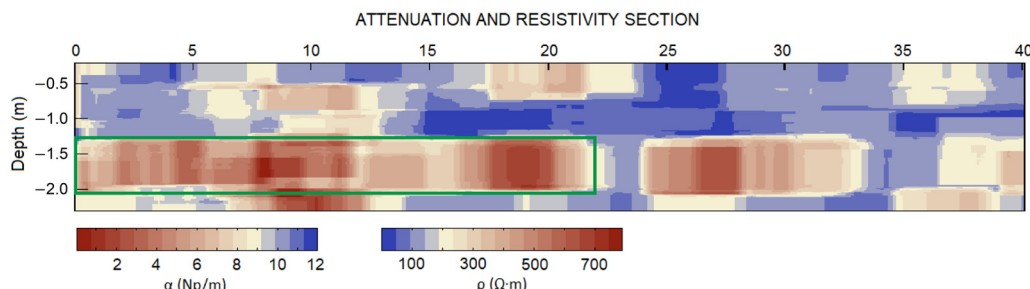

**Figure 15.** Example of a section showing the variation of attenuation coefficient and resistivity values obtained from Equation (8).

In the section shown in Figure 15, the main results regarding the resistivity distribution of subsoil represent a layer appearing at a depth of approximately 1.3 m up to 2.0 m (70 cm thickness) with higher values of resistivity than the upper and lower layers.

### 3.1. Comparison with the Tomographic Resistivity Section

For comparison, an electrical tomography profile was measured in the same area to a depth of 2.2 m. The results obtained (Figure 16) show that both the anomalies and the resistivity values of the area can be correlated. Thus, we can affirm the validity of the methodology presented in this article to obtain the resistivities from GPR data.

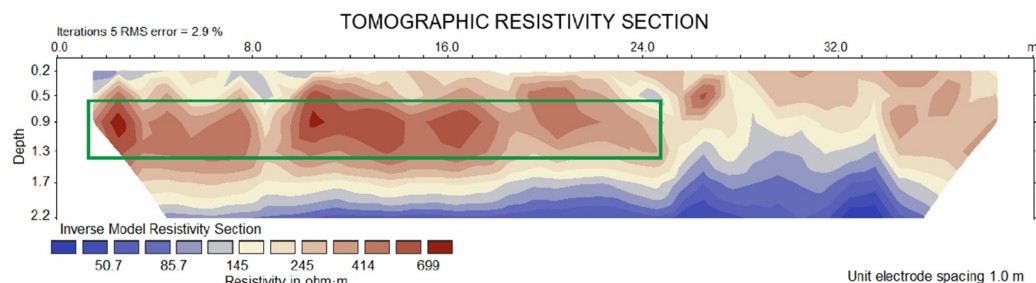

**Figure 16.** Tomographic resistivity section measured in the same area.

However, the limitations of GPR in the vertical differentiation of resistivity must be considered. This is because, within each layer considered in each trace, only the decreasing part of the envelope curve provides information about the decay produced in that layer. This is attributed to the intermediate amplitude increases mentioned above, and because the geoelectric tomography is conducted under the assumption of the continuity of resistivity values. This implies more gradual sections than those obtained from GPR, as observed in the tomographic resistivity section shown in Figure 16.

Comparing Figures 15 and 16, it is possible to point out from a qualitative point of view certain similarities that fundamentally consist of an intermediate layer in which there is an increase in resistivity values and, below this, a zone that presents lower resistivity values. The depths at which these zones appear in both sections do not coincide exactly, with the most resistive layer appearing in the section obtained from the GPR data starting at 1.3 m and in the ERT section centered at approximately 1.1 m, and the zone with lower resistivity values (blue tones), starting at 2 m in the section extracted from the GPR data and starting at 1.6 m in the ERT section.

Table 1 shows the simplified resistivity model summarized from the interpretation of the resistivity section obtained from the GPR wave attenuation study (Figure 15) and the one obtained from the ERT section.

**Table 1.** Simplified resistivity model obtained from GPR and ERT resistivity sections.

| Model—Layer | GPR | | ERT | |
|---|---|---|---|---|
| | $\rho$ ($\Omega \cdot$m) | Depth (m) | $\rho$ ($\Omega \cdot$m) | Depth (m) |
| Layer 1 | 30–100 | 0.0–1.3 | 100–200 | 0.0–0.6 |
| Layer 2 | 300–600 | 1.3–2.0 | 200–500 | 0.6–1.6 |
| Layer 3 | 100–300 | 2.0–2.2 | 30–100 | 1.6–2.2 |

Regarding the differences in the depths shown in Table 1, especially at the top and bottom of the second layer, the following should be noted. The depths assigned by ERT present their own inaccuracy due to, among other effects, the requirement of certain continuity between cells for tomographic inversion, which is not the object of this study. Regarding the depths obtained by GPR, one of the effects that can produce inaccuracy is the fact that, in this first study on the possibilities of extracting resistivity values from this technique, a single propagation velocity was used to assign the depths in the radargram. These aspects are further addressed in the discussion section.

### 3.2. Validation of the Results

In order to obtain a better description of the methodology, a flowchart was drawn (Figure 17) showing the different stages that were carried out to obtain the final result. As shown in the figure, the starting point is the original GPR data with which the different processing steps were tested. Each of these stages has had a thorough process of analysis of the partial results, which were contrasted with data from the literature and the experience of the authors, so that the final results were reliable.

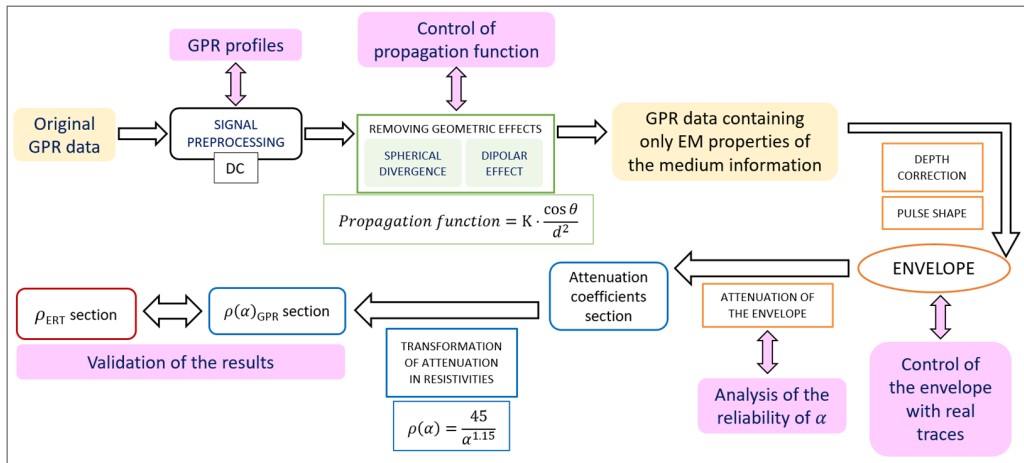

**Figure 17.** Flowchart with the main processing steps and variables involved in the methodology.

When validating the results, it must be taken into account that the resistivity of the medium, as mentioned throughout this work, is highly dependent on factors such as granulometry and the fluid that fills the pores. The granulometry determines, among other factors, the porosity and permeability of the medium. Considering also the variability of the degree of saturation of the soil and the ions in solution in the fluid, providing unequivocal lithological assignments to the resistivity values can be a source of conflict. Ultimately, and as in most geophysical methods, there is an intrinsic ambiguity that can only be resolved with the use of another geophysical method or information from mechanical boreholes or previous studies carried out in the same area.

## 4. Discussion

First, we suggest that the algorithm developed for the elimination of the continuous component (DC) by calculating the median from smaller amplitudes shows satisfactory performance. This is demonstrated by the logarithmic representation of the data before and after the application of the DC filter. Thus, in the example shown in Figure 1, the section presenting the data after the application of the DC filter shows alternating positive and negative amplitudes in similar proportions, which indicates that the DC shift has disappeared. Moreover, we indicate the advantage of the logarithmic representation, which acts as a gain filter, thereby enabling the graphical analysis to estimate the depth at which the reflection horizons can be tracked, or, the depth at which the signals can be correlated with contiguous ones. Additionally, it also helps detect significant variations in the amplitude, which can help estimate the first approximation when conducting GPR on a more or less conductive or resistive ground.

For the algorithm developed for the elimination of the considered geometric effects, namely, the flux conservation and dipole effect, this processing method needs to be applied because both effects must be compensated for to ensure that the information contained in the decay of traces refers only to the electromagnetic properties of the medium. Multiplication by the inverse of the propagation function of all the traces ensures that the geometric effects are removed from the resulting traces. In particular, this elimination simulates a situation wherein neither the spherical divergence losses nor the dipolar effect exist; however, the emitted signal would instantaneously reach its maximum intensity, and it would not decay as a function of the distance from the emitting source as it moves further away from it.

Considering the validity of this method, which will be proven if the final resistivity values are those of the medium being studied, our findings indicate that the method provides satisfactory results for signals measured with the 500 MHz frequency antenna, as traces are obtained wherein the signal of the farthest times is comparable with the first amplitudes when studying signal decays. Notably, for the study of attenuations, we

considered ratios between the amplitudes and not their absolute values. Therefore, despite reducing the strong initial undulations, no information was lost because the purpose of this study was not to consider the magnitude of amplitudes.

The velocity of the electromagnetic waves in the ground was initially unknown; however, a velocity of 0.08 m/ns was assumed; this is equivalent to a resistivity of approximately 100 $\Omega$·m, which is an average value for the type of terrain usually studied with GPR. These values are not valid for rocks wherein the wave velocity is higher. This velocity is related to Equation (5). The next step would be to apply the corresponding velocity distribution iteratively as in reflection seismic processing.

Considering the envelope function, which represents the global behavior of the signal, we highlighted the importance of conserving the zones of increasing amplitude that correspond to the appearance of interfaces (and coincide with the downward inflections of the smoothed stepped curve) if the deconvolution of the emitted signal is not conducted. The joint analysis of the smoothed and stepped curves helped establish the criteria for the automatic determination of the top and bottom of each layer and the stretches wherein the different attenuations were calculated.

We conducted a comparison between the resistivity and relative permittivity values obtained using Equation (7), with some values from the literature [35,42,64] and some from the authors' experience for standard materials, as presented in Table 2. Notably, the obtained values are within the limits defined by these authors, although in some cases, the average resistivity values can differ from those that are tabulated.

**Table 2.** Comparison of resistivity and relative permittivity values obtained from the literature [35,42,64] and those calculated from Equation (7).

| Material | Values from the Literature | | Average | Average $\varepsilon_r$ | $\varepsilon_r(\rho)$ |
|---|---|---|---|---|---|
| | $\rho$ | $\varepsilon_r$ | $\rho$ | | (Equation (7)) |
| Clays | 5-20 | 40-5 | 10 | 22 | 24.7 |
| Silts/sands | 10-1000 | 30-5 | 200 | 15 | 11.7 |
| Shales | 100-1000 | 15-5 | 500 | 10 | 9.3 |
| Limestones | 500-2000 | 8-4 | 1000 | 6 | 7.8 |
| Granite | $10^3$-$10^5$ | 6-4 | $10^4$ | 5 | 4.4 |

Notably, the attenuation and/or resistivity sections have been represented to show the lateral continuity between the correlatable resistivity values between contiguous traces by smoothing the isolines. This process facilitates the final interpretation of the sections because the traces are recorded every few centimeters. The joint representation of all sections in profiles that are often tens of meters long is not useful because of the accumulation of information in a reduced space, wherein the resolution is much higher than what can be represented. In particular, with smoothing, it is possible to unify similar resistivity values that are close to each other to determine the end areas with similar resistivity values.

Moreover, the differences in the obtained sections compared to typical electrical tomography images are also mentioned. Graphically highlighting the similarities on resistivity sections from GPR and ERT is conflicting for two reasons. Regarding GPR, the assignment of depths to the arrival times is done assuming a single velocity value for the analyzed subsoil (which is usual in GPR technique), despite the heterogeneities of the terrain that cause changes in its resistivity and therefore in its velocity. Regarding ERT, the resistivity sections do not show net changes in resistivity values, but a gradual variation, so assigning a net line of medium change is not as direct, and the assignment of depths to body contours has imprecise solutions. By way of example, consider a medium with three inhomogeneous layers, with the value of the intermediate layer higher than that of the other two. The top and bottom depths of this layer, which can be obtained from graphical inflections of the higher resistivity layer, do not match with those obtained by a 2D theoretical body model. The GPR processing proposed in this study is not intended

to replace electrical prospection techniques but to obtain additional information from the GPR profiles in relation to the resistivity distribution in the subsoil.

However, the loss of vertical resolution occurs because of the criterion used for assigning values of the attenuation coefficient. The criterion for each differentiated layer only considers the part of the envelope curve that shows decay but not the increasing part. This effect can be avoided by deconvoluting the signal emitted by the GPR antenna, which is beyond the scope of the present study. Thus, the results obtained should be considered as an estimate of the resistivity values when the media has a thickness equal to or greater than twice the distance equivalent to 5 ns of wave travel, depending on the frequency used.

Finally, we also considered the possible limitations of the described methodology in terms of its use with other frequencies. In this case, it is likely that some of the parameters set for the 500 MHz antenna, such as the propagation function constant or the one used in the depth calculation, need to be modified; however, in the latter case, the variations will not be significant.

To conclude, the proposed methodology allows a first approximation of resistivity values from GPR data, with the operational advantages of this technique. Single consecutive processing steps of the methodology should be performed iteratively by modifying the variables considered (refraction at interfaces, single velocity of the medium and variable velocity, inclination of the reflectors, etc.). In future, we intend to continue extending the code for its implementation in antennas of different frequencies and for studies in more diverse materials. This will allow its application to the rapid detection of the presence of moisture in materials such as concrete, establishing correlations between permittivity and moisture and the curing time of the concrete.

## 5. Conclusions

GPR is a widely used technique in different fields, whose most common interpretation is based on the identification of isolated reflectors or interfaces between media of different electromagnetic properties. In this work, we presented a methodology whose purpose is to extract some additional information from the GPR data in order to optimize the results obtained by this technique.

This study obtained a first approximation of the distribution of the ground resistivity values through a detailed analysis of the GPR signals, from which the values of its attenuation coefficient were extracted.

To achieve this objective, we eliminated the effects independent of the medium that caused some attenuation of the GPR waves, such as spherical divergence. The developed method was used to model the generation of GPR pulses (Figure 7), differentiating the behavior of the real pulses obtained in the GPR equipment from that of the theoretical pulse signals whose start and end are instantaneous. Accordingly, we formulated an envelope function (Figure 8) of the GPR signals to obtain signals whose attenuation corresponds only to the electrical characteristics of the traversed media. The process also established empirical relationships between the relative permittivity of granular porous media and resistivity (Figure 12), and between the attenuation coefficients and resistivity (Figure 13). The scope of these separate developments was specified in the aforementioned figures, and we concluded that these processes provide acceptable results.

In short, we obtained an estimate of the distribution of soil resistivity values, which is more reliable the greater the contrast of resistivities at the interfaces between media with different properties. In the case of low resistivity contrast or gradual changes in resistivity, this methodology may not be able to differentiate the zones because no abrupt changes will be observed in the attenuation value of the obtained curves.

Compared with other geophysical survey techniques, such as electrical tomography, whose resistivity value results are more reliable, obtaining resistivity values from GPR data has some advantages. First, GPR is considerably quicker and simpler than tomographic profiling. Second, GPR is not limited by the impossibility of drilling the ground being studied.

**Author Contributions:** Conceptualization, J.D.-C.; Data curation, L.A.-L. and B.B.; Formal analysis, J.D.-C.; Funding acquisition, B.B. and J.D.-C.; Investigation, B.B. and J.D.-C.; Methodology, L.A.-L. and J.D.-C.; Project administration, B.B.; Resources, J.D.-C.; Software, L.A.-L. and J.D.-C.; Supervision, J.D.-C.; Validation, B.B. and D.P.-P.; Visualization, L.A.-L.; Writing—original draft, L.A.-L. and J.D.-C.; Writing—review and editing, L.A.-L., B.B. and D.P.-P. All authors have read and agreed to the published version of the manuscript.

**Funding:** This research was funded by the Regional Government of Madrid, (CARESOIL-CM project grant number P2018/EMT-4317).

**Data Availability Statement:** The data presented in this study are available on request from the corresponding author. The data are not publicly available due to some special reasons.

**Conflicts of Interest:** The authors declare no conflict of interest.

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
