# Peer review of "Processing Radargrams to Obtain Resistivity Sections"

_remotesensing, doi:10.3390/rs14112639_

Round 1

Reviewer 1 Report

Dear authors, thank you for your very interesting paper. I especially appreciate the described solutions to provide a better view of the data and the removal of undesirable effects. I also want to emphasize that it was very important the fact of increasing the overall usefulness of radargrams by alloying a better data extraction from them, such as giving a gross estimation of the resistivity distribution.

My suggestions:

  • Improve the analysis of Figures 15 and 16 by only showing the same depth in both and graphically pointing out the similarities.
  • The first sentence of the 1st paragraph of chapter 2 (pp.146-148) should be revised. I couldn’t understand it.
  • Because the analysis of the GPR resistivity results VS a real resistivity section of the paper was so small, I would suggest reframing your paper about ways to improve readability of GPR radargrams and increase information extraction, rather than only focusing in resistivity.

Therefore, I propose to publish it after some minor revisions.

Reviewer 2 Report

The manuscript “Processing radargrams to obtain resistivity sections” by Lucía Arévalo-Lomas, Bárbara Biosca, David Paredes-Palacios and Jesús Díaz-Curiel presents a method to obtain electrical resistivity sections from GPR profiles.

Unfortunately, I have to suggest a rejection of the manuscript, since I think that there are methodological approaches that are in conflict with the basic theory of GPR. Moreover, the methodology is not well explained, the experimental dataset is not present and there is no information on the ERT profile that is used as comparison.

Here below, I report a series of motivations (not in order of importance) for not considering the proposed methodology as a valid and solid tool for estimating electrical resistivity from a radargram:

  1. The authors are completely ignoring an important and often dominant cause of energy loss in a GPR sounding: the reflection of part of the energy when there is a contrast of material properties. It is ignored for example at lines 141-142 and 217-219. Reflection/transmission at interfaces is a well-known cause for energy loss and is reported in all the basic geophysical books, see for example Reynolds (2011) or Davis et Annan (1989) (first citation in the manuscript).
  2. The dewow processing presented by the authors (lines 156-168) affects the amplitude of signal. You subtract a value (or a function with depth, not clear in the manuscript) to all the traces, so you reduce the amplitude of the signal in a non-linear relationship with the frequency content. A frequency high-pass filter was a better choice, because you need to reduce only the amplitude at low frequencies that are not in the range of the antenna sensitivity.
  3. Figure 1b shows a radargram with a severe ringing (I guess due to the contrast of physical properties between air and topographic surface, or another shallow interface). If this is not removed, the function energy-time (or energy depth) is affected by this bias and the real attenuation cannot be calculated. In this case an fk filter or a radon filter should be applied for preserving the original amplitude and not subtracting a background removal filter (subtraction of mean or median trace), which affect the amplitude of the recorded signal.
  4. The geometric effect is not so straightforward to correct. Figure 2 is a schematic and simplified concept, but not close to reality. The energy propagation into the ground from the transmitting antenna has a different pattern according to different antenna designs and usually is a lobe with side-lobes (see for example Daniels, 2009).
  5. “Propagation function” at lines 198-210.
    • What is the constant K and why does it have that value?
    • The propagation function is too simplified to be applied on a real case (no perfectly horizontal interfaces in natural media).
    • And there is also a wrong concept! The electromagnetic signal is reflected back to the receiver antenna only if there is a contrast in the physical properties. But this contrast modifies the propagation of the waves that are transmitted (Snell’s law), so you cannot use a straight line as propagation front.
  6. Section “Calculating the depths”. This is far too simplified and wrong. If you have reflections, you have materials with different properties, so different velocities. The assumption of a common velocity leads to wrong depths estimations. Figure 6 has a wrong concept, again, the same of point 5 above. The propagation of wave fronts is not a straight line if you have different velocities (Snell’s law). This is a basic theoretical concept that cannot be ignored.
  7. Figure 7. Is this a characteristic shape of the emitted pulse? Are the authors using a continuous wave GPR system?
  8. The envelope algorithm is not clear in the description and it looks weird in Figure 8. Why the authors are not using a Hilbert transform?
  9. Lines 281-287. The type of exponential curve is not described and this is essential in the estimation of the attenuation.
  10. The correlation between electrical resistivity and relative permittivity is material specific, it can largely depend on the ion content of the fluids and not on the type of geological medium. A general relationship between them is a strong assumption, probably wrong if not considered site-specific.
  11. It is not mentioned which kind of instruments are used both for GPR and ERT acquisitions. And where the survey has been performed, which type of geology, etc. Also lacking of essential acquisition parameters: type of array configuration used in ERT, electrode spacing, inversion; spatial and temporal sampling of GPR.
  12. At first glance, there is no correlation at all between the resistivity section from GPR (Figure 15) and the one from ERT (Figure 16), even if the comparison is not easy, since the colour scale is different for the 2 figures! So the entire approach is not verified! Why don’t test it on a synthetic dataset before applying the method on real data?
  13. The last consideration and a suggestion to the authors: if a methodology is kind of easy to implement, always think why has that approach not applied by anyone in the last 50 years? Maybe because it is not so straightforward. This is always a question that I also consider in my research.

David J. Daniels, Chapter 4 - Antennas, Editor(s): Harry M. Jol, Ground Penetrating Radar Theory and Applications, Elsevier, 2009, Pages 99-139, ISBN 9780444533487, https://doi.org/10.1016/B978-0-444-53348-7.00004-1.

Reynolds, J.M.: An Introduction to Applied and Environmental Geophysics, 2011, Wiley-Blackwell.

Reviewer 3 Report

Dear Authors,

Thank you very much for the excellent article, it is written in a very good English, is well structured, flawless and well understandable.

The article deals in detail with the basic characteristics of radar waves and the derived consequences for the data processing. The presented approach to first define internal parameters in order to improve the data quality and to remove interfering data is very complex, but is presented in a clear and comprehensible way.

Also the idea to compare the results of the GPR measurement with another method shows the scientific approach of the authors and supports the results presented in the article.

Author Response

We would like to thank this reviewer for his kind comments about our work.

Reviewer 4 Report

Dear authors, the manuscript is well-conceived and informative but it has some inadequacies.  I shall highlight them and the authors might improve the quality and readability of this research paper accordingly. 

1. The cross-cutting explanation of the methodology should be described by a schematic and conceptual flowchart.  The methodology should be described clarifying the definition of the variables and their modeling.   
2. Please include all the Software codes that have been implemented in this particular research work in the appendix section for independent simulation, testing, validation and integration.  
3. The introduction section does not provide a succinct theoretical basis for the study.  I would like to ask the authors to expand and highlight the advantages of their approaches that bring benefits to the solved issues.   
5. If possible, it will be good if the authors could add a graphical representation summarizing their results which compares controls, simulation results, all the parameters and variables directly related to ground resistivity values and attenuation coefficients.    
6. Please, briefly add future perspectives and further applied applications of this specific research work in the discussion section.  
7. The techniques and/or models presented and mentioned in the manuscript require sufficient details (including calibration, sensitivity analysis and validation) to allow other researchers to develop and test the applications later on.  Please include the parameters that I have mentioned here. More simulations and comparisons that show the advantages and the drawbacks of the proposed schema are needed.  
8. The most relevant data results should be summarized and demonstrated by a graph and a corresponding table.  
9. Please, highlight the outliers in all the tables and graphs, where relevant.  

Round 2

Reviewer 2 Report

The manuscript “Processing radargrams to obtain resistivity sections” by Lucía Arévalo-Lomas, Bárbara Biosca, David Paredes-Palacios and Jesús Díaz-Curiel presents a method to obtain electrical resistivity sections from GPR profiles.

In the first review, I pointed out several theoretical issues that make the present manuscript not acceptable for publication. Even if the authors replied and added sentences in the manuscript, several theoretical simplifications and omissions make me consider also the present version as not at the level of a scientific paper.

Here below I report several critical points:

  • From the authors’ response:

RESPONSE 1: The energy loss reported in most of literature on GPR is that of reflection / transmission at existing interfaces between media with different physical properties, which is relevant in order to analyze the corresponding parameter contrast. By the fact of in this study the energy loss in reflection / transmission at existing interfaces between media with different physical properties is not considered, it does not mean that authors ignored this effect, but it is not the analyzed effect in the GPR traces. This study aims to characterize the entire interval of signal decaying along different stretches, analyzing the decay exponent independently of contrast in its upper interface. This allows the estimation of a property in a certain medium regardless of the value of that property in the upper medium. Certainly, energy loss at existing interfaces could be analyzed if the amplitudes corresponding to maximum intensity of the actual pulses are considered, but it is not the aim of this study. To clarify this aspect pointed out by the reviewer, the last paragraph of the background section has been modified:

According to the authors’ perspective, conducting vertical electrical sounding or time-domain electromagnetic sounding in the study area was the best way to obtain an approximate resistivity value of the ground. Regarding GPR, two main energy losses of GPR signals may be considered. On one hand, a more known energy loss in reflection / transmission at existing interfaces between media with different electromagnetic properties, specially resistivity. A strong contrast in it will generate strong reflections and strong signal attenuation. On the other hand, the energy loss inside each medium due to attenuation because its no-infinite resistivity, which is the effect analyzed in this study. One main advantage of this methodology is that the attenuation calculation is mainly dependent on the resistivity values. Therefore, all decay that can be measured in the resulting trace is attributed to the absorption effects of the medium and to reflection and transmission effects at the existing interfaces owing to its electromagnetic properties.

            Comment:

The only signal recorded by a GPR are reflections. If you do not have reflections, no electromagnetic signal is recorded, so the traces are only noise. This means that you cannot ignore reflection and transmission analysing a GPR trace for attenuation.

Authors wrote:On one hand, a more known energy loss in reflection / transmission at existing interfaces between media with different electromagnetic properties, specially resistivity.. This is not true, the reflection and transmission coefficients are depending excursively on permittivity (if the material is not magnetic), not resistivity that affects the attenuation. This is again a basic knowledge in GPR, see standard books of GPR theory where the reflection coefficient is obtained from the relative permittivity of the two media at the interface.

  • From the authors’ response:

Response 3: One of the main attempt of this study is to preserve the original signal, eliminating only the geometrical effects affecting the amplitude decay. From a conventional point of view, Fig 1b shows ringing of the emitted signal, hence an fk filter or a radon filter may be applied to remove this ringing, modifying the original undulations and enhancing reflections due to variations in the subsurface. In contrast, the applied process has been designed so that it does not produce a bias with depth.

Comment:

A trace with a “ringing” cannot be analysed for attenuation. The signal at late times coming from the ringing is superimposed on the signal coming from the deeper portions of the ground and, as in this case, it shows higher amplitudes (otherwise you cannot see the “ringing”). If this is ignored, the fitting of the attenuation will give a wrong estimation of soil electrical conductivity. So, the ringing should be removed, even if the filtering may insert small artefacts.

  • From the authors’ response:

Response 5-c: In this study, the reflected wave considered is not only a unique undulation, but the corresponding to the entire emitted pulse, so that it will show the wave attenuation suffered across the transverse medium. As certainly straight line cannot be used as a propagation front excepting for planar waves, the analysis of wave propagation is made through the ray traces, which here are considered to be straight lines inside each differentiate medium. It is important to highlight that this study propose to analyze the signal decay inside each medium and not the reflections / transmissions at each interface.

Response 6-b: Regarding the use of ray trace instead the wave front, we have assumed that the ray pattern shown in Fig. 6, where the surface is parallel to the antenna, is appropriate. The reflecting wave front in each point of a reflector will be conformed following the Huygens principle and obeying the Snell’s law, so that the ray trace inside each medium will conform a straight line, between which the one arriving to the receiver dipole is represented.

Comment:

You can use straight lines inside a medium, but not straight lines crossing a boundary where you have an interface between different properties. Figure 6 is wrong! According to Snell’s law, the ray-path is deviated at the interface. This assumption is theoretically wrong and invalidate your methodology.

  • From the authors’ response:

Response 6-a: The use of a single velocity is a simplification that it is initially used to consider the analyzed propagation. In the presence of reflecting interfaces, the ray will undergo a direction change according to its velocity contrast, but the total distance traveled is a sufficiently valid approximation excepting for surface reflections. Certainly, if the proposed methodology is accepted as a GPR process to approximate the resistivity distribution in subsoil, the next step would be to apply the corresponding velocity distribution iteratively as in reflection seismic processing.

Comment:

The assumption of a homogeneous velocity is common in GPR imaging. But in the case of attenuation estimation, it cannot be ignored. As I said above, no velocity contrast, no reflections. This leads to an empty trace, so no information from the underground.

  • From the authors’ response:

Response 7: We think that the main difference between the methodological approaches of this study and the reviewer’ opinion lies on the emission system of the transmitting antenna (see Fig. 7 on characteristic shape of the emitted pulse). In this study it is considered that GPR emit short pulses with a given internal frequency (by its internal crystal responsible of this frequency) that can be stacked. Although the electronic pulse generation might generate pulses that start and finish practically instantaneous, the actual intensity that current through the transmitter antenna is not so instantaneous, but require a certain very short time to reach the generated intensity, which is showed in Fig. 7. Thus, the GPR emission is not a continuous wave GPR, but consecutive pulses with a given internal frequency.

Comment:

With the GPR system used by the authors, the pulses are repeated in time, but only to get information from different time positions. The trace cannot be acquired entirely with only one pulse. So the pulses are repeated only to sample at different times and reconstruct the entire trace. So, at the end, the signal is only 1 pulse.

  • Here below 3 examples (not even recent) of how complex is the estimation of GPR signal attenuation:

Leucci, G. (2008). Ground Penetrating Radar: The Electromagnetic Signal Attenuation and Maximum Penetration Depth. Scholarly Research Exchange, 2008, 1–7. https://doi.org/10.3814/2008/926091

Bradford, J. H. (2007). Frequency-dependent attenuation analysis of ground-penetrating radar data. Geophysics, 72(3), J7–J16. https://doi.org/10.1190/1.2710183

Irving, J. D., & Knight, R. J. (2003). Removal of wavelet dispersion from ground-penetrating radar data. 68(3), 960–970.

Reviewer 4 Report

Dear Authors, I am okay with the changes made. Thanks. 

Author Response

We would like to thank this reviewer for the changes we have made to the manuscript as a result of his/her suggestions, which we think have improved it.